# Optimal Algorithms for the Inhomogeneous Spiked Wigner model

**Justin Ko**
ENS de Lyon
justin.ko@ens-lyon.fr

**Florent Krzakala**
EPFL
florent.krzakala@epfl.ch

**Aleksandr Pak**
ENS de Lyon, EPFL
aleksandr.pak@epfl.ch

## Abstract

We study a spiked Wigner problem with an inhomogeneous noise profile. Our aim in this problem is to recover the signal passed through an inhomogeneous low-rank matrix channel. While the information-theoretic performances are well-known, we focus on the algorithmic problem. First, we derive an approximate message-passing algorithm (AMP) for the inhomogeneous problem and show that its rigorous state evolution coincides with the fixed-point equations satisfied by the Bayes-optimal estimator. Second, we deduce a simple and efficient spectral method that outperforms PCA and is shown to match the information-theoretic transition.

Low-rank information extraction from a noisy data matrix is a crucial statistical challenge. The spiked random matrix models have recently gained extensive interest in the fields of statistics, probability, and machine learning, serving as a valuable platform for exploring this issue Donoho and Johnstone [1995], Péché [2014], Baik et al. [2005]. A prominent example is the spiked Wigner model, where a rank one matrix is observed through a component-wise homogeneous noise.

Here we consider the inhomogeneous version of the model which has been recently introduced in a series of papers Barbier and Reeves [2020], Behne and Reeves [2022], Alberici et al. [2021a, 2022] and naturally arises in different contexts such as community detection Behne and Reeves [2022], deep Boltzmann machines Alberici et al. [2021b] etc. In particular, it corresponds to the dense limit of the degree-corrected stochastic block model Karrer and Newman [2011] (see the explicit mapping in Guionnet et al. [2022]). In this model the signal is observed through an inhomogeneous Gaussian noise with block variance profile. Such block-constant noise naturally arises in settings such as community detection and stochastic-block models. The assumption for the noise to be Gaussian is in no way restrictive as a universality result in Guionnet et al. [2022] shows that an entrywise transformation of data generated from a general inhomogeneous inference problem reduces any probabilistic noise to the spiked Wigner problem with Gaussian data. This generality is what makes the model appealing.

We now define the model: Consider a partition $\{1, \ldots, N\} = [N]$ into $q$ disjoint groups $C_1^N \cup \cdots \cup C_q^N = [N]$. This partition is encoded by a function $g : [N] \mapsto [q]$ which maps each index $i \in [N]$ into its group $g(i) \in [q]$. Let $\tilde{\boldsymbol{\Delta}} \in \mathbb{R}^{q \times q}$ be a symmetric matrix encoding a block-constant symmetric matrix $\boldsymbol{\Delta} \in \mathbb{R}^{N \times N}$

$$\boldsymbol{\Delta}_{ij} = \tilde{\boldsymbol{\Delta}}_{g(i)g(j)}. \tag{1}$$

We assume both the noise profile $\tilde{\boldsymbol{\Delta}}$ and the partition function $g$ to be known. This is often the case in practical applications, such as the degree-corrected block model, Karrer and Newman [2011]

37th Conference on Neural Information Processing Systems (NeurIPS 2023).

Additionally, in practical applications where the noise profile is not known, one can often empirically estimate the noise profile and assign group membership according to the empirical variances. Such an approach is likely to yield good results but lies outside the scope of the present work.

We observe the signal $\boldsymbol{x}^\star \in \mathbb{R}^N$ which is assumed to have independent identically distributed coordinates generated from some prior distribution $\mathbb{P}_0$ (i.e. $\mathbb{P}(\boldsymbol{x}^\star = \boldsymbol{x}) = \prod_{i=1}^N \mathbb{P}_0(x_i^\star = x_i)$) through noisy measurements:

$$\boldsymbol{Y} = \sqrt{\frac{1}{N}}\boldsymbol{x}^\star(\boldsymbol{x}^\star)^T + \boldsymbol{A} \odot \sqrt{\boldsymbol{\Delta}}. \tag{2}$$

Here and throughout the article $\odot$ denotes the Hadamard product, $\sqrt{\boldsymbol{\Delta}}$ is the Hadamard square-root of $\boldsymbol{\Delta}$ and $\boldsymbol{A}$ is a real-valued symmetric GOE matrix with off-diagonal elements of unit variance. The Bayes-optimal performance of this model in the asymptotic limit $N \to \infty$ was studied rigorously in Guionnet et al. [2022], Behne and Reeves [2022], Chen and Xia [2022], Chen et al. [2021] who characterized the fundamental information-theoretic limit of reconstruction in this model.

Here we focus instead on the algorithmic problem of reconstructing the (hidden) spike. **Our contributions are many-fold**:

• We show how one can construct an Approximate Message Passing (AMP) algorithm for the inhomogeneous Wigner problem, whose asymptotic performance can be tracked by a rigorous state evolution, generalizing the homogeneous version of the algorithm for low-rank factorization [Bayati and Montanari, 2011, Deshpande et al., 2015, Lesieur et al., 2017].
• We derive a fixed-point equation for AMP and show that it coincides with the fixed-point equation for the Bayes-optimal estimator obtained in Guionnet et al. [2022]
• Finally, we present a linear version of AMP [Maillard et al., 2022], that is equivalent to a spectral method. We conjecture it to be optimal in the sense that it can detect the presence of the spike in the same region as AMP. This is quite remarkable since, as shown in [Guionnet et al., 2022, Section 2.4], the standard spectral method (PCA) fails to do so.

**Related work —** The class of approximate message passing algorithms (AMP) has attracted a lot of attention in the high-dimensional statistics and machine learning community, see e.g. [Donoho et al., 2009, Bayati and Montanari, 2011, Rangan, 2011, Deshpande et al., 2015, Lesieur et al., 2017, Gerbelot and Berthier, 2021, Feng et al., 2022]. The ideas behind this algorithm have roots in physics of spin glasses Mézard et al. [1987], Bolthausen [2014], Zdeborová and Krzakala [2016]. AMP algorithms are optimal among first order methods [Celentano et al., 2020], thus their reconstruction threshold provides a bound on the algorithmic complexity in our model. Our approach to the inhomogeneous version of AMP relies on several refinements of AMP methods to handle the full complexity of the problem, notably the spatial coupling technique Krzakala et al. [2012], Donoho et al. [2013], Javanmard and Montanari [2013], Gerbelot and Berthier [2021], Rossetti and Reeves [2023].

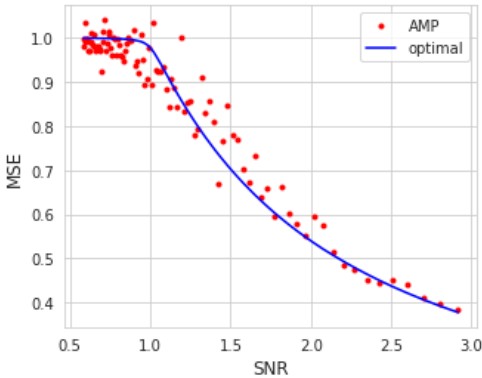

Figure 1: Performance of the inhomogeneous AMP algorithm against the information-theoretical optimal MMSE. The variance profile is proportional to $\tilde{\boldsymbol{\Delta}} = \begin{bmatrix} 1 & 3 \\ 3 & 2 \end{bmatrix}$ with two equally sized blocks with standard Gaussian prior when $N = 500$ at various snr.

Factorizing low-rank matrices is a ubiquitous problem with many applications in machine learning and statistics, ranging from sparse PCA to community detection and sub-matrix localization. Many variants of the homogeneous problem have been studied in the high-dimensional limit [Deshpande et al., 2015, Lesieur et al., 2017, Barbier et al., 2018, Lesieur et al., 2017, Alaoui et al., 2020, Lelarge and Miolane, 2019, Louart and Couillet, 2018, Barbier and Reeves, 2020]. The inhomogeneous version was discussed in detail in Guionnet et al. [2022], Behne and Reeves [2022], Alberici et al. [2021a]. Spectral methods are a very popular tool to solve rank-factorization problems [Donoho and Johnstone, 1995, Péché, 2014, Baik et al., 2005]. Using AMP as an inspiration for deriving

new spectral methods was discussed, for instance, in Saade et al. [2014], Lesieur et al. [2017], Aubin et al. [2019], Mondelli and Montanari [2018], Mondelli et al. [2022], Maillard et al. [2022], Venkataramanan et al. [2022].

# 1 Main results

**Message passing algorithm —** For each $t \geq 0$, let $(f_t^a)_{a \in [q]}$ be a collection of Lipschitz functions from $\mathbb{R} \to \mathbb{R}$, and define for $t \in \mathbb{N}$ $f_t : \mathbb{R}^N \mapsto \mathbb{R}^N$ by

$$f_t(\boldsymbol{x}) := (f_t^{g(i)}(x_i))_{i \in [N]} \in \mathbb{R}^N.$$

These functions are often called denoiser functions and can be chosen amongst several options, such as the Bayes optimal denoisers for practical applications (see Section 2), or even linear denoisers (see Section 3). We shall consider the following AMP recursion

$$\boldsymbol{x}^{t+1} = \left( \frac{1}{\sqrt{N}\boldsymbol{\Delta}} \odot \boldsymbol{Y} \right) f_t\left(\boldsymbol{x}^t\right) - \mathbf{b}_t \odot f_{t-1}\left(\boldsymbol{x}^{t-1}\right) \tag{3}$$

with the so-called Onsager term $\mathbf{b}_t = \frac{1}{N\boldsymbol{\Delta}} f_t'(x^t) \in \mathbb{R}^N$ where $\frac{1}{\boldsymbol{\Delta}}$ is the Hadamard inverse of $\boldsymbol{\Delta}$ and $f_t'$ is the vector of coordinate wise derivatives.

In practical implementations, we initialize the algorithm with some non-null $\boldsymbol{x}^0$ and let it run for a certain number of iterations. One efficient way to do this is the spectral initialization [Mondelli and Venkataramanan, 2021] with the method described in sec. 3. In Figure 1 we provide an example of the performance of the AMP together with the Bayes-optimal estimator predicted by the asymptotic theory. Even at very moderate sizes, the agreement between theory and simulation is clear.

**State evolution —** AMP was the basis of many works for the homogeneous case e.g. Deshpande et al. [2015], Lesieur et al. [2017]. Our first contribution is the introduction of an AMP and of its rigorous state evolution [Javanmard and Montanari, 2013, Theorem 1] in the inhomogeneous setting. To state a well-defined limit of the AMP, we have the following assumptions.

**Assumption 1.1.** *To ensure that our inhomogeneous AMP has a well-defined limit, we assume that*

*1. For each $a \in [q]$, we have*

$$\lim_{N \to \infty} \frac{|C_a^N|}{N} \to c_a \in (0, 1).$$

*2. For each $t \in [N]$ and $a \in [q]$, both $(f_t^a)$ and $(f_t^a)'$ are Lipschitz.*
*3. For each $a \in [q]$, there exists $\left(\sigma_a^0\right)^2 \in \mathbb{R}$ such that, in probability,*

$$\lim_{N \to \infty} \frac{1}{|C_a^N|} \sum_{i \in C_a^N} f_0^a(x_i^0) f_0^a(x_i^0) = \left(\sigma_a^0\right)^2.$$

Our first result describes the distribution of the iterates in the limit. Our mode of convergence will be with respect to $L$-pseudo-Lipschitz test functions $\phi : \mathbb{R}^M \to \mathbb{R}$ satisfying

$$|\phi(x) - \phi(y)| \leq L(1 + \|x\| + \|y\|)\|x - y\| \qquad \text{for all } x, y \in \mathbb{R}^M. \tag{4}$$

We define the following state evolution parameters $\mu_b^t$ and $\sigma_b^t$ for $b \in [q]$ through the recursion

$$\mu_b^{t+1} = \sum_{a \in [q]} \frac{c_a}{\tilde{\boldsymbol{\Delta}}_{ab}} \mathbb{E}_{x_0^\star, Z}[x_0^\star f_t^a\left(\mu_a^t x_0^\star + \sigma_a^t Z\right)] \text{ with } x_0^\star \sim \mathbb{P}_0, Z \sim \mathcal{N}(0, 1)$$

$$(\sigma_b^{t+1})^2 = \sum_{a=1}^q \frac{c_a}{\tilde{\boldsymbol{\Delta}}_{ab}} \mathbb{E}_{x_0^\star, Z} \left[ (f_t^a(\mu_a^t x_0^\star + \sigma_a^t Z))^2 \right] \text{ with } x_0^\star \sim \mathbb{P}_0, Z \sim \mathcal{N}(0, 1), \tag{5}$$

where $x_0^\star$ and $Z$ are independent. We prove that the iterates $\boldsymbol{x}_i^t$ are asymptotically equal in distribution to $\mu_{g(i)}^t x_0^\star + \sigma_{g(i)}^t Z$ where $x_0^\star \sim \mathbb{P}_0$ and $Z$ is an independent standard Gaussian.

**Theorem 1.2** (State evolution of AMP iterates in the inhomogeneous setting). *Suppose that Assumption 1.1 holds, and that $\mathbb{P}_0$ has bounded second moment. Let $\phi : \mathbb{R}^2 \to \mathbb{R}$ be a 2-pseudo-Lipschitz test functions satisfying* (4). *For any $a \in [q]$, the following limit holds almost surely*

$$\lim_{N \to \infty} \frac{1}{|C_a^N|} \sum_{i \in C_a^N} \phi(x_i^t, x_i^\star) = \mathbb{E}_{x_0^\star, Z} \phi(\mu_a^t x_0^\star + \sigma_a^t Z, x_0^\star)$$

*where $Z$ is a standard Gaussian independent from all other variables.*

**Remark 1.3.** *The notion of convergence under the L-pseudo-Lipschitz test functions induces a topology that is equivalent to the one generated by the L-Wasserstein topology [Feng et al., 2022, Remark 7.18]. We can strengthen the second moment assumption on $\mathbb{P}_0$ to finite kth moment, but the induced topology will then change to the k-Wasserstein topology, see [Javanmard and Montanari, 2013, Theorem 1].*

Even though the theoretical result above applies in the high-dimensional limit, numerical simulations show that even for medium-sized $N$ (around $500$), the behaviour of the iterates is well described by the state evolution parameters. Through the state evolution equations (5) we are able to track the iterates of the AMP iteration with just two vectors of parameters obeying the state evolution recursion: the overlap with the true signal $(\mu_a^t)_{a \in [q]}$ and its variance $(\sigma_a^t)_{a \in [q]}$. We next obtain the following necessary and sufficient condition for the overlaps of a fixed point of the iteration (2):

**Theorem 1.4** (Bayes-Optimal fixed point). *Assume AMP satisfies Assumption 1.1 and let the denoising functions be the Bayes ones* (32). *Then the overlaps $\boldsymbol{\mu} = (\mu_a)_{a \in [q]}$ in* (5) *satisfy the following fixed point equation*

$$\mu_b = \sum_{a \in [q]} \frac{c_a}{\tilde{\boldsymbol{\Delta}}_{ab}} \mathbb{E}_{x_0^\star, Z} [x_0^\star \mathbb{E}_{posterior}[x_0^\star | \mu_a x_0^\star + \sqrt{\mu_a} Z]]. \tag{6}$$

**Remark 1.5.** *The state evolution fixed point equation above coincides with the fixed point equation satisfied by the Bayes optimal estimator in [Guionnet et al., 2022, Equation 2.14].*

**A spectral method** — Given the matrix $\boldsymbol{Y}$ defined in (2) we consider the transformed matrix

$$\tilde{\boldsymbol{Y}} := \frac{\mathbb{E}_{x_0^\star}[(x_0^\star)^2]}{\sqrt{N}\boldsymbol{\Delta}} \odot \boldsymbol{Y} - \mathbb{E}_{x_0^\star}[(x_0^\star)^2]^2 \operatorname{diag}\left( \frac{1}{N\boldsymbol{\Delta}} \begin{bmatrix} 1 \\ \vdots \\ 1 \end{bmatrix} \right). \tag{7}$$

Let $\boldsymbol{c} = (c_a)_{a \in [q]}$. We define the inhomogeneous signal-to-noise (SNR) ratio of such a model by

$$\operatorname{SNR}(\boldsymbol{\Delta}) := \lambda(\boldsymbol{\Delta}) = \mathbb{E}_{x_0^\star}[(x_0^\star)^2]^2 \left\| \operatorname{diag}(\sqrt{\boldsymbol{c}}) \frac{1}{\boldsymbol{\Delta}} \operatorname{diag}(\sqrt{\boldsymbol{c}}) \right\|_{op}. \tag{8}$$

**Conjecture 1.6.** *The top eigenvalue of $\tilde{\boldsymbol{Y}}$ separates from the bulk if and only if the signal to noise ratio $\lambda(\boldsymbol{\Delta}) > 1$. In particular, if $\hat{\boldsymbol{x}}$ is the top eigenvector of $\tilde{\boldsymbol{Y}}$, then if and only if $\lambda(\boldsymbol{\Delta}) < 1$ we have:*

$$\lim_{N \to \infty} \frac{|\hat{\boldsymbol{x}} \cdot \boldsymbol{x}^\star|}{\|\hat{\boldsymbol{x}}\|\|\boldsymbol{x}^\star\|} = 0.$$

The conjecture is based on heuristic arguments identifying the fixed point of AMP with a spectral method. This matches precisely the recovery transition in [Guionnet et al., 2022, Lemma 2.15 Part (b)]. In this paper, we rigorously show that with $\operatorname{SNR}(\boldsymbol{\Delta}) < 1$ our proposed spectral method fails to recover the signal. We illustrate the eigenvalue BBP-like transition in Fig.2.

## 2 The inhomogeneous AMP algorithm

In this section, we derive the formula for the inhomogeneous AMP iteration (25). We first recall the general matrix framework of AMP from [Javanmard and Montanari, 2013]:

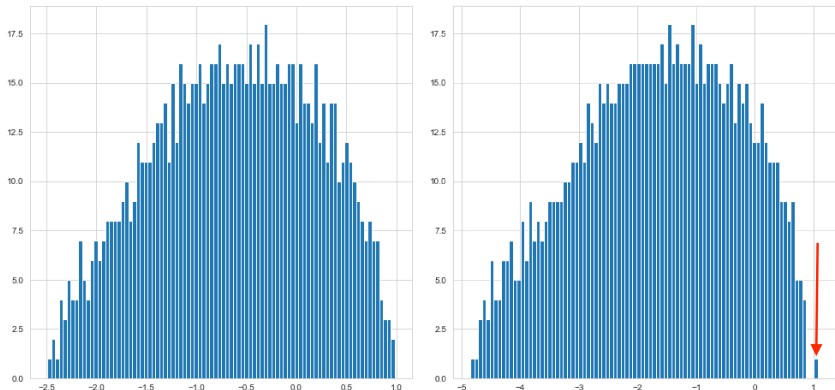

Figure 2: Illustration of the spectrum of $\tilde{\boldsymbol{Y}} \in \mathbb{R}^{10^3 \times 10^3}$ evaluated at noise profiles with snr $\lambda(\boldsymbol{\Delta}) = 0.7$ (left, before the transition) and $1.8$ (right, after the transition), with the outlying eigenvector correlated with the spike arises at eigenvalue one.

**Matrix AMP —** In the matrix setting an AMP algorithm operates on the vector space $\mathcal{V}_{q,N} \equiv (\mathbb{R}^q)^N \simeq \mathbb{R}^{N \times q}$. Each element of $\boldsymbol{v} = (v_1, \ldots, v_N) \in \mathcal{V}_{q,N}$ will be regarded as $N$-vector with entries $v_i \in \mathbb{R}^q$.

**Definition 2.1** (AMP). *A matrix AMP acting on this space is represented by $(\boldsymbol{A}, \mathcal{F}, \boldsymbol{v}^0)$, where:*

*1. $\boldsymbol{A} = \boldsymbol{G} + \boldsymbol{G}^\mathsf{T}$, where $\boldsymbol{G} \in \mathbb{R}^{N \times N}$ has iid entries $G_{ij} \sim N(0, \frac{1}{2})$.*
*2. $\mathcal{F}$ is a family of $N$ Lipschitz functions $f_t^i : \mathbb{R}^q \mapsto \mathbb{R}^q$, $i \in [N]$ indexed by time $t \in \mathbb{N}$. The family $\mathcal{F}$ encodes a function $f_t : \mathcal{V}_{q,N} \to \mathcal{V}_{q,N}$ that acts separately on each coordinate $v_j \in \mathbb{R}^q$,*

$$f_t(\boldsymbol{v}) = (f_t^1(v_1), \ldots, f_t^N(v_N)) \in \mathcal{V}_{q,N}. \tag{9}$$

*3. $\boldsymbol{v}^0 \in \mathcal{V}_{q,N}$ is a starting condition.*

The algorithm itself is a sequence of iterates generated by:

$$\boldsymbol{v}^{t+1} = \frac{\boldsymbol{A}}{\sqrt{N}} f_t\left(\boldsymbol{v}^t\right) - f_{t-1}\left(\boldsymbol{v}^{t-1}\right) \mathbf{B}_t^T \tag{10}$$

where $\mathbf{B}_t$ is the $q \times q$ Onsager matrix given by

$$\mathbf{B}_t = \frac{1}{N} \sum_{j=1}^N \partial f_t^j(\boldsymbol{v}_j^t). \tag{11}$$

where $\partial f_t^j$ denotes the Jacobian matrix of $f_t^j$. The limiting properties of the AMP sequences are well known and can be found in [Javanmard and Montanari, 2013, Theorem 1].

**The inhomogeneous AMP** — We now define an inhomogeneous AMP iteration which takes into account the block-constant structure of the noise:

**Definition 2.2.** *An inhomogeneous AMP on $\mathcal{V}_{1,N} = \mathbb{R}^N$ is represented by $(\boldsymbol{A}, \mathcal{F}, \boldsymbol{x}^0, \boldsymbol{\Delta})$, where the terms $\boldsymbol{A}, \mathcal{F}, \boldsymbol{x}^0$ are defined in Definition 2.1 and $\boldsymbol{\Delta}$ is the $N \times N$ variance profile encoded by $\tilde{\boldsymbol{\Delta}} \in \mathbb{R}^{q \times q}$ and grouping $g : [N] \to [q]$ defined by (1). We further assume that the family of functions $\mathcal{F}$ is encoded by functions $f_t^a : \mathbb{R} \mapsto \mathbb{R}$ for $a \in [q]$ which define the group dependent function*

$$f_t(\boldsymbol{x}) = (f_t^{g(1)}(x_1), \ldots, f_t^{g(N)}(x_N)) \in \mathbb{R}^N. \tag{12}$$

The sequence of iterates $\boldsymbol{x}^t \in \mathbb{R}^N$ of the $(\boldsymbol{A}, \mathcal{F}, \boldsymbol{v}^0, \boldsymbol{\Delta})$ are defined as follows:

$$\boldsymbol{x}^{t+1} = \left(\frac{1}{\sqrt{N}\sqrt{\boldsymbol{\Delta}}} \odot \boldsymbol{A}\right) f_t\left(\boldsymbol{x}^t\right) - \mathrm{b}_t \odot f_{t-1}\left(\boldsymbol{x}^{t-1}\right), \tag{13}$$

where $\frac{1}{\sqrt{\mathbf{\Delta}}}$ is the Hadamard inverse square root of the noise, and the Onsager term $\mathbf{b}_t$ has the following form

$$\mathbf{b}_t = \frac{1}{N} \begin{pmatrix} \frac{1}{\Delta_{11}}(f_t^{g(1)})'(x_1^t) & + & \cdots & + & \frac{1}{\Delta_{1N}}(f_t^{g(N)})'(x_N^t) \\ \vdots & & \vdots & & \vdots \\ \frac{1}{\Delta_{N1}}(f_t^{g(1)})'(x_1^t) & + & \cdots & + & \frac{1}{\Delta_{NN}}(f_t^{g(N)})'(x_N^t) \end{pmatrix} = \frac{1}{N\mathbf{\Delta}}f_t'(x^t) \in \mathbb{R}^N. \quad (14)$$

**State evolution of the inhomogeneous AMP —** Through a continuous embedding, we will reduce our inhomogeneous AMP to the matrix AMP framework, and recover the state evolution of the inhomogeneous AMP. We define the diagonal matrix operator blockdiag $: \mathbb{R}^N \mapsto \mathcal{V}_{q,N}$ which outputs a block diagonal matrix according to the block structure of our discretization of $[N]$:

$$\text{blockdiag}(\boldsymbol{v}) = \boldsymbol{M} \quad \text{where} \quad M_{ij} = \begin{cases} v_j & g(j) = i \\ 0 & \text{otherwise.} \end{cases}$$

Likewise, we define the projection operator blockproj $: \mathcal{V}_{q,N} \mapsto \mathbb{R}^N$ which extracts a vector of size $N$ from a $N \times q$ according to the block structure of $[N]$ by

$$\text{blockproj}(\boldsymbol{M}) = (M_{ig(i)})_{i \leq N} \in \mathbb{R}^N.$$

Under these changes of variables, we define

$$\boldsymbol{r}^t = \text{blockdiag}(\boldsymbol{x}^t) \in \mathcal{V}_{q,N} \quad \text{for } t \geq 0$$

and $\tilde{f}_t \colon (\mathbb{R}^q)^N \mapsto (\mathbb{R}^q)^N$ by

$$\left(\tilde{f}_t(\boldsymbol{r}^t)\right)_{ij} = \frac{1}{\sqrt{\tilde{\Delta}_{g(i)j}}} f_t^{g(i)}(x_i^t) \quad \text{for } i, j \in [N] \times [q]. \quad (15)$$

We encode the family of functions $\tilde{f}_t$ by $\tilde{\mathcal{F}}(\mathbf{\Delta})$.

**Lemma 2.3.** *Let $\boldsymbol{x}^t$ be iterates from the AMP $(\boldsymbol{A}, \mathcal{F}, \boldsymbol{v}^0, \mathbf{\Delta})$. Then the iterates $\boldsymbol{r}^t := \text{blockdiag}(\boldsymbol{x}^t)$ follow the generalized matrix AMP $(\boldsymbol{A}, \tilde{\mathcal{F}}(\mathbf{\Delta}), \boldsymbol{r}^0)$.*

*Proof.* We will show that the projection of the iterates $\boldsymbol{r}^t$ from $(\boldsymbol{A}, \tilde{\mathcal{F}}(\mathbf{\Delta}), \boldsymbol{r}^0)$ are the iterates from $(\boldsymbol{A}, \mathcal{F}, \boldsymbol{v}^0, \mathbf{\Delta})$. It is easy to check that

$$\frac{\boldsymbol{A}}{\sqrt{N}\sqrt{\mathbf{\Delta}}} f_t(\boldsymbol{x}^t) = \text{blockproj}\left(\frac{\boldsymbol{A}}{\sqrt{N}} \tilde{f}_t(\boldsymbol{r}^t)\right). \quad (16)$$

Next, notice that Jacobian is given by

$$\partial \tilde{f}_t^i(\boldsymbol{r}_i^t) = \begin{bmatrix} 0 & \cdots & \frac{1}{\sqrt{\tilde{\Delta}_{g(i)1}}}(f_t^{g(i)})'(x_i^t) & \cdots & 0 \\ \vdots & \ddots & \vdots & \ddots & \vdots \\ 0 & \cdots & \frac{1}{\sqrt{\tilde{\Delta}_{g(i)q}}}(f_t^{g(i)})'(x_i^t) & \cdots & 0 \end{bmatrix},$$

which is a matrix where only the column number $g(i)$ has non-zero elements. Applying (11), we thus get for $a, b \in [q] \times [q]$

$$(\boldsymbol{B}_t)_{ab} = \frac{1}{N\sqrt{\tilde{\mathbf{\Delta}}_{ab}}} \sum_{i:g(i)=a} (f_t^a)'(x_i^t). \quad (17)$$

It follows that

$$\mathbf{b}_t \odot f_{t-1}(\boldsymbol{x}^{t-1}) = \text{blockproj}\left(\tilde{f}_{t-1}(\boldsymbol{r}^{t-1})\boldsymbol{B}_t^T\right).$$

$\square$

As a consequence, the inhomogeneous state evolution equations in Theorem 1.2 follow immediately from the state evolution equations of $(\boldsymbol{A}, \tilde{\mathcal{F}}(\boldsymbol{\Delta}), \boldsymbol{r}^0)$ discussed in [Javanmard and Montanari, 2013, Section 2.1]. This follows from the observation that given the law of $\boldsymbol{r}^t$ in the high dimensional limit, the law of $\boldsymbol{x}^t = \mathrm{blockproj}(\boldsymbol{r}^t)$ is straightforward to compute. We define

$$(\sigma_b^{t+1})^2 := \sum_{a=1}^{q} \frac{c_a}{\tilde{\boldsymbol{\Delta}}_{ab}} \mathbb{E}\left[(f_t^b(Z_b^t))^2\right]. \tag{18}$$

We will show that the distribution of the iterate $x_i^t$ is asymptotically normal with mean 0 and variance $(\sigma_{g(i)}^t)^2$.

**Lemma 2.4** (Behavior of the inhomogeneous AMP iterates). *Suppose that Assumption 1.1 holds, and that $\mathbb{P}_0$ has a bounded second moment. Let $\phi : \mathbb{R}^2 \to \mathbb{R}$ be a L-pseudo-Lipschitz test function satisfying (4). For any $a \in [q]$, then the following limit holds almost surely*

$$\lim_{N \to \infty} \frac{1}{|C_a^N|} \sum_{i \in C_a^N} \phi(x_i^t, x_i^\star) = \mathbb{E}_{x_0^\star, Z} \phi(\sigma_a^t Z, x_0^\star)$$

*where $Z$ is an independent standard Gaussian.*

*Proof.* In matrix-AMP [Javanmard and Montanari, 2013, Th. 1], the marginals of the iterates $\boldsymbol{r}^t$ from $(\boldsymbol{A}, \tilde{\mathcal{F}}(\boldsymbol{\Delta}), \boldsymbol{r}^0)$ are approximately Gaussian and encoded by the positive definite matrices

$$\widehat{\boldsymbol{\Sigma}}_a^t = \mathbb{E}\left[\tilde{f}_t^i\left(\boldsymbol{Z}^t\right) \tilde{f}_t^i\left(\boldsymbol{Z}^t\right)^\top\right] \qquad \text{for all } i \in C_a^N, \tag{19}$$

$$\text{with } \boldsymbol{Z}^t \sim N(0, \boldsymbol{\Sigma}^t) \text{ and } \Sigma^{t+1} = \sum_{a=1}^{q} c_a \widehat{\Sigma}_a^t. \tag{20}$$

We now show that $a \in [q]$ and $i \in C_a^N$, $\widehat{\boldsymbol{\Sigma}}_a^t$ depends only on $(\sigma_a^t)^2 = \boldsymbol{\Sigma}_{aa}^t$. Indeed, by the definition of $\tilde{f}_t^i$ we have

$$\widehat{\boldsymbol{\Sigma}}_a^t(k, l) = \mathbb{E}\left[(\tilde{f}_t^i\left(\boldsymbol{Z}^t\right))_k (\tilde{f}_t^i\left(\boldsymbol{Z}^t\right))_l\right] = \mathbb{E}\left[\frac{1}{\sqrt{\tilde{\boldsymbol{\Delta}}_{ak}}} \frac{1}{\sqrt{\tilde{\boldsymbol{\Delta}}_{al}}} (f_t^a(Z_a^t))^2\right], \tag{21}$$

where $Z_a^t \sim N(0, (\sigma_a^t)^2)$ is the $a$th component of the Gaussian vector $\boldsymbol{Z}_t$. The key observation here is that by construction our function $\tilde{f}_t^i, \mathbb{R}^q \mapsto \mathbb{R}^q$ depends only on the ith component $Z_i^t$ of the Gaussian vector $\boldsymbol{Z}^t$. To characterize the limiting distribution of $\boldsymbol{x}^t = \mathrm{blockproj}(\boldsymbol{r}^t)$, we only need to keep track of the variances $(\sigma_j^t)^2, j \in [N]$. Using (21), for a given $a \in [q]$ and $i \in C_a^N$ we write

$$\widehat{\boldsymbol{\Sigma}}_a^t(g(j), g(j)) = \mathbb{E}\left[(\tilde{f}_t^i\left(\boldsymbol{Z}^t\right))_{g(j)} (\tilde{f}_t^i\left(\boldsymbol{Z}^t\right))_{g(j)}\right] = \frac{1}{\tilde{\boldsymbol{\Delta}}_{ag(j)}} \mathbb{E}\left[(f_t^i(Z_i^t))^2\right]. \tag{22}$$

Finally, with (20) we get that for any $b \in [q]$ and any $j \in C_b^N$, using $Z_a^t \sim N(0, (\sigma_a^t)^2)$,

$$(\sigma_j^{t+1})^2 = (\sigma_b^{t+1})^2 = \boldsymbol{\Sigma}_{bb}^{t+1} = \sum_{a=1}^{q} c_a \widehat{\boldsymbol{\Sigma}}_a^t(g(j), g(j)) = \sum_{a=1}^{q} \frac{c_a}{\tilde{\boldsymbol{\Delta}}_{ag(j)}} \mathbb{E}\left[(f_t^a(Z_a^t))^2\right] \tag{23}$$

$\square$

**The inhomogeneous spiked Wigner model in the light of the AMP approach —** We now generalize the state evolution equations from Lemma 2.4 to spiked matrices with an inhomogenous noise profile as was stated in Theorem 1.2. This reduction via a change of variables is standard, see for example [Deshpande et al., 2015, Lemma 4.4]. Remember that in the inhomogeneous version of the spiked Wigner model we observe the signal $\boldsymbol{x}^\star$ through an inhomogeneous channel:

$$\boldsymbol{Y} = \sqrt{\frac{1}{N}} \boldsymbol{x}^\star (\boldsymbol{x}^\star)^T + \boldsymbol{A} \odot \sqrt{\boldsymbol{\Delta}}. \tag{24}$$

Our AMP algorithm is defined with the following recursion:

$$\boldsymbol{x}^{t+1} = \left(\frac{1}{\sqrt{N}\boldsymbol{\Delta}} \odot \boldsymbol{Y}\right) f_t\left(\boldsymbol{x}^t\right) - \mathbf{b}_t \odot f_{t-1}\left(\boldsymbol{x}^{t-1}\right) \tag{25}$$

where $\mathbf{b}_t = \frac{1}{N\boldsymbol{\Delta}} f_t'(\boldsymbol{x}^t)$ and $f$ is encoded by the family of functions in Definition 2.2. The main difference in contrast to the iteration (13) is that our data matrix $\boldsymbol{Y}$ is no longer a centered matrix, while $\frac{1}{\sqrt{\boldsymbol{\Delta}}} \odot \boldsymbol{A}$ is. We would like to reduce (25) to an iteration of the form (13) with respect to a different parameter $\boldsymbol{s}^t$ which is uniquely determined by $\boldsymbol{x}^t$

$$\boldsymbol{s}^{t+1} = \left(\frac{1}{\sqrt{N}\sqrt{\boldsymbol{\Delta}}} \odot \boldsymbol{A}\right) f_t\left(\boldsymbol{s}^t\right) - \mathbf{b}_t \odot f_{t-1}\left(\boldsymbol{s}^{t-1}\right).$$

Doing so will allow us to recover the limiting laws of the iterates from Lemma 2.4. This is done via a standard change of variables to recenter $\boldsymbol{Y}$. We will sketch the argument in this section, and defer the full proof of Theorem 1.2 to the Appendix A in the Supplementary Material.

To simplify notation, let us denote $f_t(\boldsymbol{x}^t) := \hat{\boldsymbol{x}}^t$. We proceed following the approach of [Deshpande et al., 2015, Lemma 4.4]. We rewrite (25) using the definition of Y to get

$$\begin{aligned}
\boldsymbol{x}^{t+1} &= \left(\frac{1}{\sqrt{N}\boldsymbol{\Delta}} \odot \boldsymbol{Y}\right)\hat{\boldsymbol{x}}^t - \mathbf{b}_t \odot \hat{\boldsymbol{x}}^{t-1} \\
&= \left(\frac{1}{N\boldsymbol{\Delta}} \odot \boldsymbol{x}^\star(\boldsymbol{x}^\star)^{\boldsymbol{T}}\right)\hat{\boldsymbol{x}}^t + \left(\frac{1}{\sqrt{N}\sqrt{\boldsymbol{\Delta}}} \odot \boldsymbol{A}\right)\hat{\boldsymbol{x}}^t - \mathbf{b}_t \odot \hat{\boldsymbol{x}}^{t-1}.
\end{aligned} \tag{26}$$

If indices are independent, then by the strong law of large numbers one would expect that

$$\left(\left(\frac{1}{N\boldsymbol{\Delta}} \odot \boldsymbol{x}^\star(\boldsymbol{x}^\star)^{\boldsymbol{T}}\right)\hat{\boldsymbol{x}}^t\right)_j = x_j^\star \sum_{a\in[q]} \sum_{i\in C_a^N} \frac{1}{N} \frac{x_i^\star \hat{x}_i^t}{\boldsymbol{\Delta}_{ji}} \to x_j^\star \sum_{a\in[q]} \frac{c_a}{\boldsymbol{\Delta}_{ji_a}} \mathbb{E}[x_0^\star \hat{x}_{i_a}^t], \tag{27}$$

where $i_a$ is some index belonging to the group $C_a^N$ and $x_0^\star$ is a random variable distributed according to the prior distribution $\mathbb{P}_0$. For $b \in [q]$ and $i \in C_b^N$, we define the block overlap $\mu_b^t$ using the recursion

$$\mu_i^{t+1} = \mu_b^{t+1} = \sum_{a\in[q]} \frac{c_a}{\tilde{\boldsymbol{\Delta}}_{ab}} \mathbb{E}_{x_0^\star, Z}[x_0^\star f_t^a\left(\mu_a^t x_0^\star + \sigma_a^t Z\right)], \tag{28}$$

where $Z$ is a standard Gaussian random variable independent from all others sources of randomness. Notice that (28) is precisely the asymptotic behavior of the summation appearing in (27).

We now make a change of variables and track the iterates

$$\boldsymbol{s}^0 = \boldsymbol{x}^0 - \boldsymbol{\mu}^0 \odot \boldsymbol{x}^\star \qquad \boldsymbol{s}^t = \boldsymbol{x}^t - \boldsymbol{\mu}^t \odot \boldsymbol{x}^\star, \quad t \geq 1 \tag{29}$$

where $\boldsymbol{\mu}^0$ is the vector of block overlaps of the initial condition $\boldsymbol{x}^0$ with the truth. We reduced the (25) iteration to the following iteration in which we easily recognize a version of (13):

$$\boldsymbol{s}^{t+1} = \left(\frac{1}{\sqrt{N}\sqrt{\boldsymbol{\Delta}}} \odot \boldsymbol{A}\right) f_t\left(\boldsymbol{s}^t + \boldsymbol{\mu}^t \odot \boldsymbol{x}^\star\right) - \mathbf{b}_t \odot f_{t-1}\left(\boldsymbol{s}^{t-1} + \boldsymbol{\mu}^{t-1} \odot \boldsymbol{x}^\star\right) \tag{30}$$

with the initial condition $\boldsymbol{s}^0 = \boldsymbol{x}^0 - \boldsymbol{\mu}^0 \odot \boldsymbol{x}^\star$ and the Onsager term taken from (14) is given by

$$\mathbf{b}_t = \frac{1}{N\boldsymbol{\Delta}} f_t'(\boldsymbol{s}^t + \boldsymbol{\mu}^t \odot \boldsymbol{x}^\star). \tag{31}$$

Using Lemma 2.4, we can recover the asymptotic behavior of the iterates $\boldsymbol{x}^t$ given in (25) by computing the iterates $\boldsymbol{s}^t + \boldsymbol{\mu}_t \odot \boldsymbol{x}^\star$ where $\boldsymbol{s}^t$ follows (30) and $\boldsymbol{\mu}_t$ satisfies (28). From this reduction we obtain the following state evolution equations describing the behaviour of (25):

1. $x_j^t \cong \mu_{g(j)}^t x_0^\star + \sigma_{g(j)}^t Z$ for $j \in [N]$, where $Z \sim \mathcal{N}(0,1)$
2. $\mu_b^{t+1} = \sum_{a\in[q]} \frac{c_a}{\tilde{\boldsymbol{\Delta}}_{ab}} \mathbb{E}_{x_0^\star, Z}[x_0^\star f_t^a\left(\mu_a^t x_0^\star + \sigma_a^t Z\right)]$ with $x_0^\star \sim \mathbb{P}_0, Z \sim \mathcal{N}(0,1)$
3. $(\sigma_b^{t+1})^2 = \sum_{a=1}^q \frac{c_a}{\tilde{\boldsymbol{\Delta}}_{ab}} \mathbb{E}_{x_0^\star, Z}\left[(f_t^a(\mu_a^t x_0^\star + \sigma_a^t Z))^2\right]$ with $x_0^\star \sim \mathbb{P}_0, Z \sim \mathcal{N}(0,1)$.

This (informally) characterizes the limiting distribution of the state evolution of the iterates from the inhomogeneous AMP stated in Theorem 1.2.

**Fixed-point equation of state evolution in the Bayes-optimal setting —** Suppose that we know the prior distribution $\mathbb{P}_0$ of $x_0^\star$. The Bayes-optimal choice for the denoising functions $f_t^j, j \in [N]$ is simply the expectation of $x_0^\star$ with respect to the posterior distribution,

$$f_t^j(r) = f_t^{g(j)}(r) = \mathbb{E}_{posterior}[x_0^\star | \mu_{g(j)}^t x_0^\star + \sigma_{g(j)}^t Z = r]. \tag{32}$$

Under this Bayes-optimal setting, we can simplify the equations obtained in the previous section and see that AMP estimator is indeed an optimal one by studying its fixed point.

*Proof of Theorem 1.4.* For this choice of $f_t^j$ the Nishimori identity (see for example [Lelarge and Miolane, 2019, Proposition 16]) states that for $a \in [q]$ and $j \in C_a^N$,

$$\tilde{\mu}_a^t := \mathbb{E}_{x_0^\star, Z}[x_0^\star f_t^a(\mu_a^t x_0^\star + \sigma_a^t Z)] = \mathbb{E}\left[(f_t^a(\mu_a^t x_0^\star + \sigma_a^t Z))^2\right]. \tag{33}$$

In this setting, the state evolution equations from Theorem 1.2 reduce to

$$\begin{cases} \tilde{\mu}_a^t = \mathbb{E}_{x_0^\star, Z}[x_0^\star f_t^a(\mu_a^t x_0^\star + \sigma_a^t Z)] \\ \mu_b^{t+1} = \sum_{a \in [q]} \frac{c_a}{\tilde{\Delta}_{ab}} \tilde{\mu}_a^t, b \in [q] \\ (\sigma_b^{t+1})^2 = \sum_{a \in [q]} \frac{c_a}{\tilde{\Delta}_{ab}} \tilde{\mu}_a^t, b \in [q]. \end{cases} \tag{34}$$

Remarkably with the Bayes-optimal choice of the denoising functions we have that for $t \geq 1$ for each block $b \in [q]$, $\mu_b^{t+1} = (\sigma_b^{t+1})^2$. Therefore a necessary and sufficient condition for an estimator to be a fixed point of the state evolution is to simply have its overlaps $\mu_b^t$ unchanged by an iteration of the state evolution. This translates into the following equation for the overlaps $\mu_b, b \in [q]$

$$\mu_b = \sum_{a \in [q]} \frac{c_a}{\tilde{\Delta}_{ab}} \mathbb{E}_{x_0^\star, Z}[x_0^\star \mathbb{E}_{posterior}[x_0^\star | \mu_a x_0^\star + \sqrt{\mu_a} Z]]. \tag{35}$$

The result of Theorem 1.4 now follows immediately. $\qquad\square$

## 3 A spectral method adapted to the inhomogeneous spiked Wigner model

**From AMP to a spectral method —** Remarkably, AMP and the state evolution machinery associated with it can help us design a simple spectral algorithm that matches the information-theoretic phase transition [Guionnet et al., 2022, Remark 2.16]. Recall that Theorem 1.2 does not require the denoising functions $f_t$ to be Bayes-optimal, but can be applied to any Lipschitz family of functions. In this section, we analyze the state evolution for the family of identity functions, $f_t(x) = x$. We can gain some intuition motivating such a choice by considering simple priors, such as the Rademacher prior. In the case of the Rademacher prior a simple computation shows that the Bayes-optimal choice of the denoising functions yields $f_t^j(\star) = tanh(\star)$. In the first order approximation we have $tanh(x) \approx x$. Thus, at least for the Rademacher prior, the choice of identity functions as denoising functions corresponds to the first order approximation of the Bayes-optimal choice. By Remark 3.1, we can assume that the entries of $\boldsymbol{x}^\star$ have unit variance. With this choice of denoising functions the AMP iteration will simply become:

$$\boldsymbol{x}^{t+1} = \left(\frac{1}{\sqrt{N}\boldsymbol{\Delta}} \odot \boldsymbol{Y}\right) \boldsymbol{x}^t - \mathbf{b}_t \odot \boldsymbol{x}^{t-1} \quad \text{where} \quad \mathbf{b}_t = \frac{1}{N\boldsymbol{\Delta}} f_t' = \frac{1}{N\boldsymbol{\Delta}} \begin{bmatrix} 1 \\ \vdots \\ 1 \end{bmatrix}. \tag{36}$$

If we denote $\boldsymbol{B}_t = \operatorname{diag}(\mathbf{b}_t)$, it is easy to see that the fixed point of this iteration yields

$$\boldsymbol{x} = \left(\frac{1}{\sqrt{N}\boldsymbol{\Delta}} \odot \boldsymbol{Y}\right) \boldsymbol{x} - \boldsymbol{B}_t \boldsymbol{x} \tag{37}$$

so any $\boldsymbol{x}$ fixed by the AMP iteration (36) must be an eigenvector of the matrix

$$\tilde{\boldsymbol{Y}} = \left(\frac{1}{\sqrt{N}\boldsymbol{\Delta}} \odot \boldsymbol{Y}\right) - \boldsymbol{B}_t = \left(\frac{1}{\sqrt{N}\boldsymbol{\Delta}} \odot \boldsymbol{Y}\right) - \operatorname{diag}\left(\frac{1}{N\boldsymbol{\Delta}} \begin{bmatrix} 1 \\ \vdots \\ 1 \end{bmatrix}\right). \tag{38}$$

A simple spectral method consists in taking the principal eigenvector (associated to the largest eigenvalue) of the matrix $\tilde{\boldsymbol{Y}} = \frac{\boldsymbol{Y}}{\sqrt{N}\boldsymbol{\Delta}} - \boldsymbol{B}_t$.

**Analysis of the spectral method using state evolution —** The spectral algorithm described above behaves like the AMP iteration with identity denoising functions around its fixed point. Therefore we can analyze this spectral algorithm using state evolution machinery for the AMP iteration. In the case of identity functions we have $f_t^a(x) = x$ for all $a \in [q]$, so

$$\mathbb{E}_{x_0^\star, Z}[x_0^\star f_t^a(\mu_a^t x_0^\star + \sigma_a^t Z)] = \mathbb{E}_{x_0^\star, Z}[x_0^\star(\mu_a^t x_0^\star + \sigma_a^t Z)] = \mu_a^t \tag{39}$$

$$\mathbb{E}_{x_0^\star, Z}[(f_t^a(\mu_a^t x_0^\star + \sigma_a^t Z))^2] = \mathbb{E}_{x_0^\star, Z}[(\mu_a^t x_0^\star + \sigma_a^t Z)^2] = (\mu_a^t)^2 + (\sigma_a^t)^2 \tag{40}$$

which transforms state evolution equations (5) into the following simple form:

$$\mu_b^{t+1} = \sum_{a \in [q]} \frac{c_a}{\Delta_{ba}} \mu_a^t \quad \text{and} \quad (\sigma_b^{t+1})^2 = \sum_{a \in [q]} \frac{c_a}{\Delta_{ba}} ((\mu_a^t)^2 + (\sigma_a^t)^2). \tag{41}$$

Rewriting the overlap state evolution in a matrix form we get for $\boldsymbol{c} = (c_a)_{a \in [q]}$ that

$$\operatorname{diag}(\sqrt{\boldsymbol{c}})\boldsymbol{\mu}^{t+1} = \operatorname{diag}(\sqrt{\boldsymbol{c}})\frac{1}{\boldsymbol{\Delta}}\operatorname{diag}(\sqrt{\boldsymbol{c}})\left(\operatorname{diag}(\sqrt{\boldsymbol{c}})\boldsymbol{\mu}^t\right). \tag{42}$$

The special form of state evolution above is rather informative and we can interpret it as follows. First of all, in the regime where $\lambda(\boldsymbol{\Delta}) = \left\|\operatorname{diag}(\sqrt{\boldsymbol{c}})\frac{1}{\boldsymbol{\Delta}}\operatorname{diag}(\sqrt{\boldsymbol{c}})\right\|_{op} < 1$, we can see from (42) that any iteration of the AMP recursion (36) contracts the vector overlap (multiplying it by a matrix with the operator norm smaller than 1). Thus the only possible overlap of a fixed point of (36) is 0. Moreover, we have defined the matrix $\tilde{\boldsymbol{Y}}$ in a way that any eigenvector of this matrix is a fixed point of the AMP recursion (36). Thus any eigenvector of $\tilde{\boldsymbol{Y}}$ must have zero expected overlap with the signal, meaning that the spectral method is uninformative in the regime $\lambda(\boldsymbol{\Delta}) < 1$.

Second of all, even though we cannot say anything rigorous in the regime $\lambda(\boldsymbol{\Delta}) > 1$, looking at state evolution equations of the AMP recursion (36), we see instability (overlap blows up but so does the variance). We conjecture that the principal eigenvector of $\tilde{\boldsymbol{Y}}$ correlates with the signal exactly in the regime $\lambda(\boldsymbol{\Delta}) > 1$ which would imply that the BBP transition happens precisely at $\lambda(\boldsymbol{\Delta}) = 1$.

**Remark 3.1.** *In general, we can let $\gamma = \mathbb{E}_{x_0^\star}[(x_0^\star)^2]$ and consider the normalized matrix*

$$\bar{\boldsymbol{Y}} = \frac{\boldsymbol{Y}}{\gamma} = \sqrt{\frac{1}{N}}\frac{\boldsymbol{x}^\star(\boldsymbol{x}^\star)^T}{\gamma} + \boldsymbol{A} \odot \frac{\sqrt{\boldsymbol{\Delta}}}{\gamma} = \sqrt{\frac{1}{N}}\bar{\boldsymbol{x}}^\star(\bar{\boldsymbol{x}}^\star)^T + \boldsymbol{A} \odot \sqrt{\bar{\boldsymbol{\Delta}}}$$

*for $\bar{\boldsymbol{x}} = \frac{\boldsymbol{x}}{\sqrt{\gamma}}$ and $\bar{\boldsymbol{\Delta}} = \frac{\boldsymbol{\Delta}}{\gamma^2}$. Notice that the entries of $\bar{\boldsymbol{x}}$ now have unit variance. Under this setting, the transition of the transformation in (38) applied to $\bar{Y}$, which appears in (7), has transition at*

$$\lambda(\bar{\boldsymbol{\Delta}}) = \left\|\operatorname{diag}(\sqrt{\boldsymbol{c}})\frac{1}{\bar{\boldsymbol{\Delta}}}\operatorname{diag}(\sqrt{\boldsymbol{c}})\right\|_{op} = \mathbb{E}_{x_0^\star}[(x_0^\star)^2]^2\left\|\operatorname{diag}(\sqrt{\boldsymbol{c}})\frac{1}{\boldsymbol{\Delta}}\operatorname{diag}(\sqrt{\boldsymbol{c}})\right\|_{op} < 1$$

*which is the generalized SNR defined in (8).*

## Acknowledgments and Disclosure of Funding

We thank Alice Guionnet and Lenka Zdeborova for valuable discussions. We acknowledge funding ´ from the ERC Project LDRAM: ERC-2019-ADG Project 884584, and by the Swiss National Science Foundation grant SNFS OperaGOST, 200021 200390

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

# A  Proof of Theorem 1.2

We now provide a rigorous proof of the result that was sketched in Section 2. This proof is essentially identical to [Deshpande et al., 2015, Lemma 4.4]. Recall the iterates (30) given by $s^0 = x^0 - \mu^0 \odot x^\star$ and

$$s^{t+1} = \left( \frac{1}{\sqrt{N}\sqrt{\Delta}} \odot A \right) f_t \left( s^t + \mu^t \odot x^\star \right) - \mathbf{b}_t \odot f_{t-1} \left( s^{t-1} + \mu^{t-1} \odot x^\star \right), \qquad (43)$$

where $\mu^t = (\mu_i^t)_{i \leq N}$ is given by the recursion

$$\mu_i^{t+1} = \mu_{g(i)}^{t+1} = \sum_{a \in [q]} \frac{c_a}{\tilde{\Delta}_{ag(i)}} \mathbb{E}_{x_0^\star, Z}[x_0^\star f_t^a \left( \mu_a^t x_0^\star + \sigma_a^t Z \right)]$$

and $x^\star = (x_i^\star)_{i \in [N]}$ is a vector with independent coordinates distributed according to $\mathbb{P}_0$. By Lemma 2.4, for each $a \in [q]$, and any pseudo-Lipschitz function $\phi : \mathbb{R} \to \mathbb{R} \mapsto \mathbb{R}$ we have that almost surely

$$\lim_{N \to \infty} \frac{1}{|C_a^N|} \sum_{i \in C_a^N} \phi(s_i^t, x_i^\star) = \mathbb{E}_{x_0^\star, Z} \phi(\sigma_a^t Z, x_0^\star) \qquad (44)$$

where

$$(\sigma_b^{t+1})^2 := \sum_{a=1}^q \frac{c_a}{\tilde{\Delta}_{ab}} \mathbb{E}_Z \left[ (f_t^b(Z_b^t))^2 \right].$$

as was defined in (18). For any pseudo-Lipschitz function $\psi : \mathbb{R} \to \mathbb{R}$, we have $\phi(x, y) = \phi(x - \mu_a^t y)$ is also pseudo-Lipschitz, so (44) implies that

$$\lim_{N \to \infty} \frac{1}{|C_a^N|} \sum_{i \in C_a^N} \psi(s_i^t + \mu_a^t x_i^\star) = \mathbb{E}_{x_0^\star, Z} \psi(\sigma_a^t Z + \mu_a^t x_0^\star) \qquad (45)$$

almost surely.

Now let $x^t$ be the iterates from the spiked AMP iteration for the inhomogeneous Wigner matrix (24) we derived in (26)

$$x^{t+1} = \left( \frac{1}{N\Delta} \odot x^\star (x^\star)^T \right) \hat{x}^t + \left( \frac{1}{\sqrt{N}\sqrt{\Delta}} \odot A \right) \hat{x}^t - \mathbf{b}_t \odot \hat{x}^{t-1}. \qquad (46)$$

It now suffices to show that for fixed $t$ and all $a \in [q]$ that

$$\lim_{N \to \infty} \frac{1}{|C_a^N|} \sum_{i \in C_a^N} (\psi(s_i^t + \mu_a^t x_i^\star) - \psi(x_i^t)) = 0 \qquad (47)$$

almost surely. This will imply that $s_i^t + \mu_a^t x_i^\star$ and $x_i^t$ have the same asymptotic distribution which finish the proof of Theorem 1.2 by (45).

We now prove (47). Since $\psi$ is $L$-pseudo-Lipschitz we have

$$|\psi(s_i^t + \mu_a^t x_i^\star) - \psi(x_i^t)| \leq L(1 + |s_i^t + \mu_a^t x_i^\star| + |x_i^t|)|s_i^t + \mu_a^t x_i^\star - x_i^t|$$
$$\leq 2L|s_i^t + \mu_a^t x_i^\star - x_i^t|(1 + |s_i^t + \mu_a^t x_i^\star| + |s_i^t + \mu_a^t x_i^\star - x_i^t|).$$

The Cauchy–Schwarz inequality implies that

$$\left| \frac{1}{|C_a^N|} \sum_{i \in C_a^N} (\psi(s_i^t + \mu_a^t x_i^\star) - \psi(x_i^t)) \right|$$

$$\leq \frac{2L}{C_a^N} (\sqrt{C_a^N} \|s_a^t + \mu_a^t x_a^\star - x_a^t\|_2 + \|s_a^t + \mu_a^t x_a^\star\|_2 \|s_a^t + \mu_a^t x_a^\star - x_a^t\|_2 + \|s_a^t + \mu_a^t x_a^\star - x_a^t\|_2^2)$$

where $s_a^t = (s_i^t)_{i \in C_a^N} \in \mathbb{R}^{|C_a^N|}$, $x_a^t = (x_i^t)_{i \in C_a^N} \in \mathbb{R}^{|C_a^N|}$. Therefore, to prove (47) it suffices to prove that for all $t \geq 0$,

$$\lim_{N \to \infty} \frac{1}{|C_a^N|} \|s_a^t + \mu_a^t x_a^\star - x_a^t\|_2^2 \to 0 \qquad (48)$$

$$\limsup_{N \to \infty} \frac{1}{|C_a^N|} \|s_a^t + \mu_a^t x_a^\star\|_2^2 \to 0 \qquad (49)$$

Clearly, if we initialize $\boldsymbol{x}^0$, $\boldsymbol{s}^0$ at zero then (48) and (49) are satisfied by our state evolution equations (5). Notice that (49) follows directly from (45) applied to the square function. We use here that we assumed that the second moment of $x^\star$ is finite.

We now focus on proving (48) through strong induction. By definition of the iterates (43) and (46),

$$
\begin{aligned}
(\boldsymbol{s}_a^t &+ \mu_a^t \boldsymbol{x}_a^\star - \boldsymbol{x}_a^t) \\
&= \Bigg[ \left( \frac{1}{\sqrt{N}\sqrt{\boldsymbol{\Delta}}} \odot \boldsymbol{A} \right) f_{t-1}\left( \boldsymbol{s}^{t-1} + \boldsymbol{\mu}^{t-1} \odot \boldsymbol{x}^\star \right) - \left( \frac{1}{\sqrt{N}\sqrt{\boldsymbol{\Delta}}} \odot \boldsymbol{A} \right) f_{t-1}(\boldsymbol{x}^{t-1}) \\
&\quad + \boldsymbol{\mu}^t \odot \boldsymbol{x}^\star - \left( \frac{1}{N\boldsymbol{\Delta}} \odot \boldsymbol{x}^\star (\boldsymbol{x}^\star)^T \right) f_{t-1}(\boldsymbol{x}^{t-1}) \\
&\quad + \mathbf{b}_{t-1}^x \odot f_{t-2}(\boldsymbol{x}^{t-2}) - \mathbf{b}_{t-1}^s \odot f_{t-2}\left( \boldsymbol{s}^{t-2} + \boldsymbol{\mu}^{t-2} \odot \boldsymbol{x}^\star \right) \Bigg]_{i \in C_a^N}
\end{aligned}
$$

where $[\cdot]_i$ corresponds to the $i$th row of a vector and $\mathbf{b}_{t-1}^x$ and $\mathbf{b}_t^s$ are the Onsager terms defined in (14) with respect to $\boldsymbol{x}^{t-1}$ and $\boldsymbol{s}^{t-1}$ respectively. The Cauchy–Schwarz inequality and Jensen's inequality imply that there exists some universal constant $C$ such that

$$
\begin{aligned}
\frac{1}{|C_a^N|} &\| \boldsymbol{s}_a^t + \mu_a^t \boldsymbol{x}_a^\star - \boldsymbol{x}_a^t \|_2^2 \\
&\leq \frac{C}{|C_a^N|} \sum_{i \in C_a^N} \frac{1}{N} \left\| \left[ \frac{1}{\sqrt{\boldsymbol{\Delta}}} \odot \boldsymbol{A} \right]_i \right\|_2^2 \| [f_{t-1}\left( \boldsymbol{s}^{t-1} + \boldsymbol{\mu}^{t-1} \odot \boldsymbol{x}^\star \right) - f_{t-1}(\boldsymbol{x}^{t-1})]_i \|_2^2 \\
&\quad + \frac{C}{|C_a^N|} \sum_{i \in C_a^N} \left( \mu_a^t - \left[ \frac{1}{N\boldsymbol{\Delta}} (f_{t-1}(\boldsymbol{x}^{t-1}) \odot \boldsymbol{x}^\star) \right]_i \right)^2 (x_i^\star)^2 \\
&\quad + \frac{C}{|C_a^N|} \sum_{i \in C_a^N} ([\mathbf{b}_{t-1}^x]_i - [\mathbf{b}_{t-1}^s]_i)^2 [f_{t-2}(\boldsymbol{s}^{t-2} + \boldsymbol{\mu}^{t-2} \odot \boldsymbol{x}^\star)]_i^2 \\
&\quad + \frac{C}{|C_a^N|} \sum_{i \in C_a^N} [\mathbf{b}_{t-1}^x]_i^2 ([f_{t-2}(\boldsymbol{x}^{t-2})]_i - [f_{t-2}(\boldsymbol{s}^{t-2} + \boldsymbol{\mu}^{t-2} \odot \boldsymbol{x}^\star)]_i)^2
\end{aligned}
$$

We now control each term separately.

1. To control the first term, notice that the matrix $\frac{1}{N} \left[ \frac{1}{\sqrt{\boldsymbol{\Delta}}} \odot \boldsymbol{A} \right]$ has iid entries within blocks and the sizes of the blocks diverge with the dimension, so we can control the sums of the squares of within each block using standard operator norm bounds Anderson et al. [2010]. The first term vanishes in the limit because $f$ is pseudo-Lipschitz so we can apply the induction hypothesis bound which controls (48) at time $t-1$.

2. To control the second term, notice that for $i \in C_a^N$ by Lemma 2.4 applied to the pseudo-Lipschitz function $y f_{t-1}(x)$ that

$$
\left[ \frac{1}{N\boldsymbol{\Delta}} (f_{t-1}(\boldsymbol{x}^{t-1}) \odot \boldsymbol{x}^\star) \right]_i \to \mu_a
$$

almost surely. This implies that the average of such terms vanishes since we assumed that the second moment $\mathbb{E}[x_0^\star]^2$ is finite.

3. To control the third and fourth terms, we can expand the definition of the Onsager terms and use the assumption that $f'$ is pseudo-Lipschitz and almost surely bounded. Both terms vanish because our strong induction hypothesis gives us control of (48) at time $t-2$.

Since all terms vanish in the limit, we have proven (48) for all $a \in [q]$, which finishes the proof of statement (47) and the proof of Theorem 1.2.

## B   Comparison with a naive PCA spectral method

In this appendix, we wish to show how the spectral method we propose differs, in practice, from a naive PCA. We provide an example of the spectrums of $\boldsymbol{Y}$ and $\tilde{\boldsymbol{Y}}$ before and after the transition at

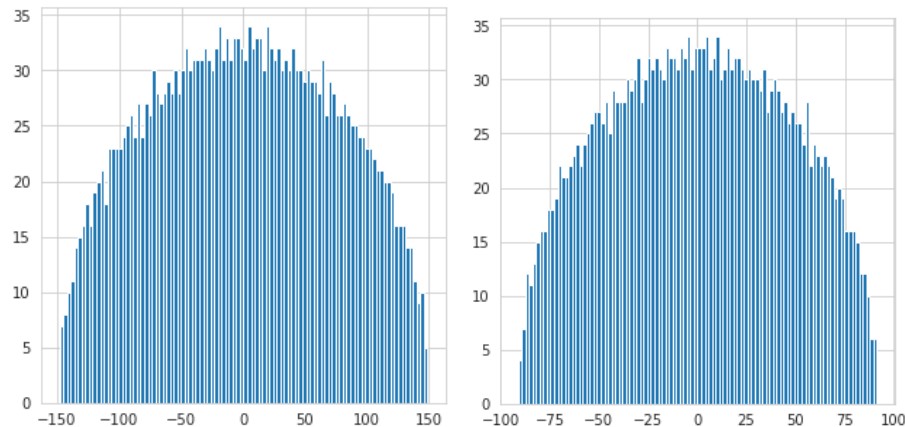

Figure 3: Illustration of the spectrum of $Y \in \mathbb{R}^{2500 \times 2500}$ evaluated at noise profiles with snr $\lambda(\mathbf{\Delta}) = 0.7$ (left, before the transition) and on the left and $1.8$ on the right (after the transition). There is no outlying eigenvalue in contrast to the transformed matrix: the transition for a naive spectral method is sub-optimal.

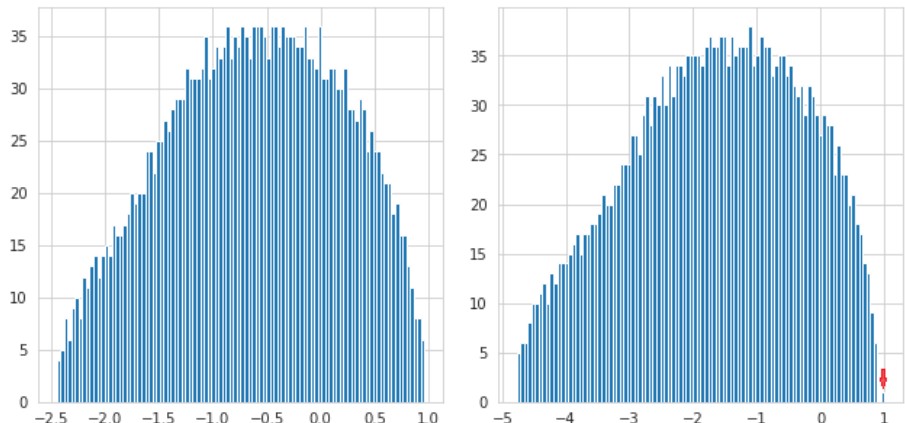

Figure 4: Illustration of the spectrum of $\tilde{Y} \in \mathbb{R}^{2500 \times 2500}$ evaluated at noise profiles with snr $\lambda(\mathbf{\Delta}) = 0.7$ (left, before the transition) and on the left and $1.8$ on the right (after the transition), with the outlying eigenvector correlated with the spike arises at eigenvalue one. This is at variance with the results of the naive method in Fig.3

$\mathrm{SNR}(\Delta) = 1$. In Figure 3 there is no clear separation of the extremal eigenvalue of $Y$ from the bulk around this transition. This is in contrast to Figure 4 where there is an extremal eigenvalue of $\tilde{Y}$ appearing at value one.

