# OpenReview forum: "Optimal Algorithms for the Inhomogeneous Spiked Wigner Model"
_NeurIPS.cc/2023/Conference — NeurIPS 2023 poster_

### Official Review · Reviewer_ifZx · 2023-06-20

**Soundness:** 3 good
**Presentation:** 2 fair
**Contribution:** 3 good
**Rating:** 6
**Confidence:** 4

**Summary:**

This paper considers approximate message passing algorithms for reconstructing a rank-1 signal when corrupted by a symmetric matrix of noise with a block-variance structure; it is assumed the signal _x^*_ has iid coordinates generated from a prior distribution.  One then forms the matrix $Y = x^* (x^*)^T/\sqrt{N} + A \odot \sqrt{\Delta}$, where $\Delta$ is a blockwise constant, positive matrix.  The matrix $A$ is a symmetric gaussian matrix with iid off-diagonal entries of size $N \times N$.

The number of blocks $q$ appears to bounded independent of matrix size, and it is assumed one has access to the scale matrix Delta.

The first algorithm is a proper generalization of an AMP recursion for the case of a rank-1 signal with Wigner noise.  The state evolution is shown to converge (Theorem 1.2 -- using a modification of existing techniques) and correspond to the solution of the Bayes optimal estimator (Theorem 1.4).

The spectral method designs a linear recurrence which is (locally? and conditionally?) optimal, in that it also recovers the same estimator by computing a principal eigenvector of an associated matrix.

**Strengths:**

1) The mathematical content is sound.  The main content (theorem 1.2) is proven in the supplemental material.  This adds an algorithmic aspect to a model which has attracted recent information theoretic attention.
2) The paper presents an algorithm which is optimal for the problem posed.  The algorithm is part of a larger well-studied class.
3) The performance of the algorithm is illustrated numerically in a simple case.
4) A linear method is presented, which may reproduce the more complicated general AMP performance for the hidden spike.

**Weaknesses:**

0) The article does not provide any broader context for the technical results it develops; all but the first few pages are technicalities related to AMP theory.  It is targeted at experts in approximate message passing, and it does not develop much of the information-theoretic aspects of the model (which I gather are proven in Guionnet et al).  There is no conclusion.  There is nothing in the way of practical considerations or relations to application (although I would say this alone is forgivable, if the paper were otherwise immaculate).  Much of the main text is occupied by technicalities related to the formulation of the main theorem and in summoning relevant AMP theory from Javanmard and Montanari as well as Deshpande et al.
1) The main theorem (1.2) is an adaptation of an existing result.  Moreover, approximate message passing algorithms are well studied and many theorems exist for them.  The presence of non-iid noise makes it somewhat unique, but I think it is fair to say that this is not a big extension of existing theorems.  (In particular, there is a change of variables to connect the homogeneous and inhomogeneous cases).
2) The spectral method, which is introduced, is largely left half-baked.  There is a conjecture (1.6) related to it, and there is a 1 page description of how the method is developed.  There is an equation (42) showing that the overlap evolution is unstable when a certain Perron-Frobenius eigenvalue is larger than 1.  But finally, the main points here are left as conjectures.

**Questions:**

Major:

0) Much of the valuable main paper space is used on technicalities of how one AMP is formally recast as another.  What conclusions should be drawn from this?  What is the context of the information-theoretic properties of the model?  What is the surprise of the linear method (mentioned briefly in the third bullet at the start?).  Lots of needed context is missing from this paper.
1) The spectral method proposed at the end would seem to suggest that there is, finally, no need for the AMP algorithm (spectral methods are wildly simpler).  However the actual claims about what the spectral method does are left quite implicit (reading section 3).  In simple language, what do you conjecture happens for the spectral method above/below what transition?  A simple theorem in section 3 might be appropriate.
2) What is the recovery transition, and why should the spectral method be optimal?  (or what more than optimism justifies Conjecture 1.6)?  Good numerical results would be appropriate.
3) The version of the AMP algorithm that solves the inhomogeneous spiked Wigner problem is finally presented in (32).  The assumptions that you formulate are not clearly directly relevant to this theorem (i.e. you could formulate your algorithm without all the heavy assumptions).  This means that in some sense these assumptions are really technicalities suited for those who want AMP details and/or proof details.  What are the minimal assumptions you need for actually solving the inhomogeneous spiked Wigner model?
4) A discussion of the extent to which it is possible to actually implement the AMP iteration for the posterior would add to the broader appeal.

Minor:
* What is t in Assumption 1.1 part 2.  Does the family include all t?  Also this sentence is malformed.  What is the assumption?
* What is M in (4)?
* Eq (11) contains an $x_j$ as yet unused?

**Limitations:**

The assumptions are clear.

---

> ### Author Rebuttal · Authors · 2023-08-09
>
> We express our gratitude to the reviewer for their insightful comments and valuable suggestions. We will incorporate all the suggestions into the final version, regardless of acceptance.
>
> • Weakness 0: The main motivation of this paper was to approach the analysis of an inhomogeneous spiked Wigner model from an algorithmic point of view to complement the already extensive informational theoretical analysis of such models in Barbier and Reeves (2020); Behne and Reeves (2022); Alberici et al. (2022a,b); Guionnet et al. (2022). Through this process, we discovered that a linearization of the AMP algorithm implied that a simple spectral method applied on a transformed matrix will achieve weak recovery. The weak recovery transition of this spectral method is conjectured to coincide with the information theoretic transition for weak recovery.
>
> • Weakness 1: While our proof uses a reduction to a known theorem, we fail to see how it is a weakness. By connecting with an existing framework, existing algorithms can also be simply extended to the inhomogeneous framework.
>
> Additionally, the derivation of the inhomogeneous AMP in (3.5) is a technical non-trivial contribution as it involves a balanced normalization of the matrix by the variance, which is not an obvious choice. A more natural normalization by the standard deviation was proven to be suboptimal in Guionnet et al. (2022), and applying the classical AMP for the spiked Wigner model for this matrix led to very poor results. Recasting the AMP provided a simple proof in the end, but finding the correct embedding essentially required us to derive an AMP from scratch. Furthermore, local analysis of the inhomogeneous AMP led to an interesting novel conjecture in random matrix theory, which states that the spectral method applied to a transformed matrix in (2.5) achieves the BBP transition at the optimal SNR.
>
> • Weakness 2: While the final result is left as a conjecture, it rests on solid and time-tested heuristic methods (see e.g. Maillard, Krzakala, Lu and Zdeborova [2021], or Venkataramanan, Kogler, and Mondelli [2022]) and is backed by numerical simulations.
>
> A direct rigorous treatment of this spectral method is a hard open problem in random matrix theory. Tools have been developed for matrices with noise profiles, but very few explicit formulas exist in comparison to Wigner matrices. More intuitive guesses for an optimal matrix were proposed in Guionnet et al. [2022], but were proven to be incorrect. In general, characterizations of outlying eigenvalues are given implicitly for inhomogeneous matrices, so guessing the correct optimal matrix is completely non-trivial. The linearization of the fixed point of an AMP in this work led us to recover the optimal matrix (7). Our intuition gained from the AMP is a significant step towards a rigorous treatment of the spectral method, that will likely require a very fine analysis.
>
> • Question 0: a) The model has been introduced (and its information-theoretic properties characterized) in Guionnet et al. [2022], Behne and Reeves [2022], Chen et al. [2021], Alberici et al [2022] (see line 34-36), motivated by the inhomogenous version of low-rank factorization problems (see line 14-24)
>
> b) The surprise for the linear method lies in the fact that existing methods, such as PCA or reweighted PCA, fail in this and other inhomogeneous models, see e.g. Guionnet et al. [2022]. Finding a successful spectral method was an open question.
>
> Our algorithmic point of view was used to tackle a peculiarity that was seen in the information theoretic recovery phase transitions. In Guionnet et al. [2022], an intuitive ”homogenization” of the noise resulted in a BBP phase transition at a non-optimal signal to noise ratio. We wanted to use approximate message passing and an analysis of a linearization around the fixed points of this AMP to find the correct way to transform the noise profile, which is defined on equation (7). Conjecture 1.6 on the optimality of the spectral method is also a nice conjecture in RMT that can be solved by studying a difficult system of quadratic vector equations. Our algorithmic point of view provided a nice conjecture that is supported by the stability and convergence of numerical simulations.
>
> • Question 1: We agree that the spectral method proposed at the end is a remarkable approach, and certainly simpler than AMP. However, there is no way to guess the form of the operator without deriving AMP in the first place! The derivation of AMP and its linearization to obtain the spectral method is our contribution. The spectral method is conjectured to achieve weak recovery of the prior in the theoretical detectability regime. The full AMP algorithm is shown to approach the optimal MMSE.
>
> • Question 2: In this model, the information theory analysis indicates that no recovery of the hidden truth is possible (no matter the algorithm) as long as SNR < 1. If SNR > 1, however, it is possible to recover a noisy version of the hidden truth. This is called ”weak recovery” in information theory, see Guionnet et al. [2022], Behne and Reeves [2022]. Our AMP is shown to be (asymptotically) able to perform weak recovery in practice (and in fact recover the MMSE estimator in linear time, a task that in principle requires sampling a high-dimensional distribution).
>
> • Question 3: We present these conditions for the mathematically minded audience. In practice, it suffices to use either spectral or random initializations to achieve good results numerically.
>
> • Question 4: AMP can be easily implemented in python, see for example the github repositories by Takashi Takahashi or Kuan Hsieh. Our AMP is very close to the spatially coupled AMP that is being used in error-correcting-codes, see e.g. Barbier, & Krzakala. (2017) or Barbier; Dia; Macris (2019).

---

> > ### Comment · Reviewer_ifZx · 2023-08-11
> >
> > Thanks for the clarifications.
> >
> > I think I was unfairly pessimistic in my initial review score -- especially ignoring the role the AMP theorem played in deriving the spectral algorithm -- and I have increased the score to 6.    The paper contains plenty for a NeurIPS publication: a theorem establishing a rigorous AMP convergence for a natural inferential problem and a derivative (in the non-mathematical sense) spectral algorithm which is clearly nontrivial and is a natural candidate for follow-up work.

---

### Official Review · Reviewer_qRVY · 2023-06-25

**Soundness:** 2 fair
**Presentation:** 2 fair
**Contribution:** 2 fair
**Rating:** 5
**Confidence:** 4

**Summary:**

This paper studies the (symmetric) rank-1 matrix estimation problem with inhomogeneous noise. Here inhomogeneous noise refers to a symmetric noise matrix that is block-wise constant where the number of blocks is a constant relative to the dimension.
This paper proposes an approximate message passing (AMP) algorithm and shows the corresponding state evolution result.
Another piece of contribution is the design of a spectral algorithm that outputs the principal eigenvector of a rescaled and recentered matrix.
Numerics suggest that this outperforms the naive estimator of the principal eigenvector of the data matrix per se.

**Strengths:**

The most interesting (at least to me) part of the paper is Section 3 where a nonstandard spectral algorithm is introduced and analyzed to some extent.
As the authors commented, this estimator outperforms (at least numerically) the naive one corresponding to the original matrix Y.

Another satisfactory aspect of the result is the coincidence between the fixed point of AMP and that of the Bayes-optimal estimator (i.e., E[x^* | Y]), though this is not surprising.

**Weaknesses:**

1. The majority of the paper is devoted to AMP and its state evolution whose proof is a rather standard reduction to the matrix-valued AMP by Javanmard--Montanari. I didn't check the details carefully since everything goes as expected. But it's still good to see things written down formally.

2. Section 3 is interesting at a heuristic level. However, I have a doubt regarding the authors' claim.

In line 131, it is claimed that "we rigorously show that with SNR<1 our proposed spectral method fails to recover the signal". I don't think the analysis in Section 3 constitutes a proof of this claim.
It was shown that the trivial fixed point of an AMP with linear denoiser is attractive when SNR<1. I agree with this, but this does not imply that the asymptotic overlap of the spectral estimator is 0 when SNR<1.
The iterate of the linearized AMP converges (in constant number of steps) to the principal eigenvector (i.e., the spectral estimator) only when a spectral gap is present.
When SNR<1, there is no spectral gap and it is unclear how the iterate of linearized AMP is related to the principal eigenvector.
It may converge to some other vector, depending on the initialization.
In fact, rigorously speaking, I think it's fundamentally unlikely to prove subcritical behaviour by exploiting linearized AMP.
The analysis only proves the attraction of 0 when SNR<1, which is an *evidence* that the phase transition threshold is 1.
However, this implies neither "spectral fails when SNR<1" nor "spectral works when SNR>1".

3. If I understand correctly, the Delta matrix is assumed to be *known*. A very important aspect that was not discussed at all (correct me if I'm wrong) is what happens when Delta is unknown which appears (to me) to be a slightly more realistic assumption.
In that case the Bayes-AMP is no longer a practical algorithm (even with warm start) and the spectral algorithm is also not computable.
In fact, is it fair to say that the proposed spectral algorithm outperforms the naive one *because* it uses Delta information?
Can the price of lacking the knowledge of Delta be quantified?
I know this may go beyond the scope of the present paper.
But it seems to be an interesting nontrivial point that's worth mentioning/discussing.

**Questions:**

1. Line 7, "info-theoretic optimal Bayes fixed-point equations" sounds like a confusing phrase. Consider expanding it as "the FP equations satisfied by the Bayes-optimal estimator".

1. Line 17, replace "etc..." with "etc."

2. Line 20 "block-constant Gaussian noise" is mildly confusing (sorry for being pedantic). The noise is never constant; consider saying "Gaussian noise with block-constant variance profile" or something like that.

3. Line 46, is the word "detect" properly used? Or do the authors mean "weak recovery"? My understanding is that they are not the same question.

3. I couldn't parse the grammar of 2. of Assumption 1.1.

5. L in the definition of PL functions seems to be a floating parameter. In standard theory, L is tied to the regularity of prior distributions and initializers. This needs to be quantified.

5. There are multiple mistakes in Remark 1.3. The correct statement should be "convergence under all L-PL functions is equivalent to convergence in Wasserstein-L". Also, bounds on k-th (k>2) moment is a *stronger* assumption than bounds on 2nd moment, consequently, convergence in W_k is *stronger* than W_2.

5. In Equation (6), has the notation E_{posterior} been defined? If not, then this notation is definitely too sloppy.

5. Line 137, there are redundant spaces in "N - vector".

5. Line 141, please explicitly write i in [N] when introducing the notation f_t^i.

5. Equation (33), I'm not familiar with the physics jargon "Nishimori identity" and didn't check Lelarge--Miolane, but isn't this just the law of total expectation (https://en.wikipedia.org/wiki/Law_of_total_expectation)?

5. In Section 3, an AMP with denoiser f_t(x) = x is considered. Why is identity denoiser a good thing to consider, besides its simplicity? I understand this comes from linearization. But for the sake of being self-contained, I suggest comment on this.

4. This paper handles heterogeneous noise. Is it true that with essentially no additional effort, heterogeneous prior can also be handled?
At least when the partition for the prior is the same as that for the noise?
I didn't check the details and may be wrong.

5. The number of blocks q is assumed to be a constant relative to N. What happens if it grows with N? Clearly if q=N nothing can be said. But this question makes sense for sufficiently slowly growing q. After the reduction to matrix-valued AMP, the SE result of Javanmard--Montanari is no longer applicable since they require the width of the matrix iterates to be a constant. Could the authors comment on what can potentially happen here?

---

> ### Author Rebuttal · Authors · 2023-08-09
>
> We express our gratitude to the reviewer for their insightful comments and valuable suggestions. We will incorporate all the suggestions into the final version, regardless of acceptance.
>
> Weakness 2: From the information theoretical analysis for these models, no recovery of the hidden truth is possible (no matter the algorithm) when the SNR $< 1$. We have shown that when the SNR $< 1$ any fixed point of the AMP recursion must have zero expected overlap with the true signal. One way of seeing it is that for SNR $< 1$ state evolution implies that any iteration of the AMP recursion contracts the vector overlap (it is being multiplied by a matrix with the operator norm smaller than 1), thus the only possible overlap of a fixed point is 0. Since any eigenvector of the matrix $\tilde{\boldsymbol{Y}}$ obtained by linearizing AMP is a fixed point of AMP, it must also have zero expected overlap with the signal. This subtlety should be further explained in Section 3.
>
> We believe indeed that there are subtle points for a fully rigorous AMP proof of the spectral method performance. We have only shown that the linear AMP is uninformative in this regime, but the performance of the spectral method still remains open. We are confident, however, that we have recovered the right matrix to analyze for such a spectral approach because it is based on a mathematically solid framework (see e.g. Maillard, Krzakala, Lu and Zdeborova [2021], or Venkataramanan, Kogler, and Mondelli [2022]) and backed by numerical simulations. A rigorous argument would require fine control of the convergence of the linear AMP, see e.g. High Dimensional Robust M-Estimation: Asymptotic Variance via Approximate Message Passing by Montanari and Donoho (2013) or the more recent, Learning Gaussian Mixtures with Generalised Linear Models: Precise Asymptotics in High-dimensions by Loureiro et al.) Alternatively, a direct rigorous treatment of the spectral method directly from random matrix theory is a challenging but interesting open problem.
>
> Weakness 3: Indeed the matrix $\tilde{\Delta}$ and the partition function $g$ are assumed to be known. Note, however, that this does not have to be a limitation.
>
> In the degree-corrected block model Karrer and Newman (2011), for instance, the noise profile is perfectly known and given by the degrees of nodes in the graph. Similarly , for practical applications, one can empirically estimate the variance profile and assign group membership according to the empirical variances. We can first estimate both $\tilde{\Delta}$ and $g$ without knowing either of them in advance. Then we can work with the estimators rather than with the true quantities. An interesting direction for future work would be to see how AMP which uses estimators instead of the true $\tilde{\Delta}$ and $g$ performs and quantify how the lack of information impacts the performance of the algorithm, for instance using the Expectation Maximization framework in AMP (e.g. as in Expectation-Maximization Gaussian-Mixture Approximate Message Passing,  Vila \& Schniter 2013).
>
> Question 1: This indeed would be a better phrasing.
>
> Question 2: Fixed.
>
> Question 3: Gaussian noise with block-constant variance profile is a better term.
>
> Question 4: Yes, detect in the context of line 46 refers to weak recovery
>
> Question 5: Line 99 should be replaced by: "For each $t \in [N], a \in [q]$" $f_{t}^{a}$ and $(f_{t}^{a})^{\prime}$ are Lipschitz."
>
> Question 6: Theorem 1 should be stated to apply to $2$--Pseudo Lipschitz functions, but can be generalized to a larger class of functions if we assume more moments on the prior. We have corrected this.
>
> Question 7: Fixed.
>
> Question 8: The notation $\mathbb{E}_{\text{posterior}}$ stands for the expected value with the respect to the posterior distribution.
>
> Question 9: Fixed.
>
> Question 10: We will add $i \in [N]$.
>
> Question 11: Nishimori identity is indeed just a clever rewriting of the law of total expectation.
>
> Question 12: We can gain some intuition motivating the choice $f_{t}(x) = x$ by considering simple priors, such as the Rademacher prior. In the case of the Rademacher prior a simple computation shows that the Bayes-optimal choice of the denoising functions yields $f_{t}^{j}(\star) = tanh(\star)$. In the first order approximation we have $tanh(x) \approx x$. Thus, at least for the Rademacher prior, the choice of identity functions as denoising functions corresponds to the first order approximation of the Bayes-optimal choice.
>
> Question 13: Heterogeneous prior can be handled as long as the coordinates are independent and the partition for the prior corresponds to the block structure of the variance profile. Although for the Bayes-optimal choice of denoising functions $f_{t}^{j}$ in the AMP different priors would certainly impact the calculation of the posterior mean in the following definition (eq. 32 of the article) as long as the prior stays the same within a block, the denoising function $f_{t}^{j} = f_{t}^{g(j)}, t \in [N], j \in [q]$ also remains the same within a given block.
>
> Question 14: This is a very interesting comment, but it is also unknown what happens for these models when the number of blocks varies depending on $N$ algorithmically nor from the information theoretic point of view. One obstacle is that the fixed point equations for the optimal estimators in the large $N$ limit depends on the $q$, so making sense of even classifying an optimal estimator is non-trivial in this setting. It remains a nice problem to explore in the future.

---

> > ### Comment · Reviewer_qRVY · 2023-08-13
> > **Thank you for the prompt and detailed reply**
> >
> > I thank the authors for their prompt and detailed reply (and revision of the manuscript which I didn't check).
> >
> > "Weakness" 2: I agree with the authors' reply and I'm fine with heuristics (as long as it's made clear which is rigorous which is heuristic).
> > After all, these heuristics are likely to yield accurate predictions.
> > If space permitted, please consider making more clear in the manuscript the gap between attraction of trivial fixed point and the ineffectiveness of spectral estimator (as discussed in the reply).
> >
> > Weakness 3: I agree that in some settings, Delta is naturally known.
> > Regarding the proposed strategy of first estimating \tilde{Delta} & partition then plugging them into AMP (or whatever subsequent procedure), is it obvious that in the proportional regime these can be estimated consistently? If not, then there is an asymptotic price to pay. Another point is that we observe Y which contains the signal. For the purpose of estimating the nuance parameters, the signal is an interference. One model that one can play with is that in addition to Y, we also observe Z which is pure noise (for the purpose of estimating tilde{Delta} and g).
> > A spiritually similar model is proposed and studied here: https://arxiv.org/abs/2211.00986.
> > In any case, this goes well beyond the scope of this paper.
> > My only point here is that the assumption that tilde{Delta} and g are known deserves a few sentences of remark.
> >
> > Q12: This motivation makes sense. If space permitted, please consider adding a line or two discussing the choice of linear denoiser.
> >
> > Q14: Thanks for the reply. I agree that this in general looks difficult.
> >
> > Overall, my evaluation remains the same -- this paper is sufficiently interesting for NeurIPS.

---

### Official Review · Reviewer_kH5e · 2023-07-07

**Soundness:** 3 good
**Presentation:** 3 good
**Contribution:** 3 good
**Rating:** 7
**Confidence:** 2

**Summary:**

The paper provides an analysis of an AMP algorithm for the spiked Wigner model with inhomogeneous noise. The paper builds on the matrix AMP framework to derive the state evolution equations for the considered AMP recursion for the studied model. The paper further shows that if the denoising functions are the Bayes one, then the fixed point equation of the state evolution of the AMP algorithm is the same as the one satisfied by the Bayes optimal estimator.

The paper also leverages the developed machinery to study the properties of a spectral algorithm which is motivated by considering the the identity denoising functions. It is conjectured that this spectral algorithm exhibits optimal phase transition.

**Strengths:**

I find the paper to be generally well-written and it is not hard to follow. The problem that is considered is interesting and the presented results generalize previously known results for the spiked Wigner model with homogenous noise to the non-homogenous case.

**Weaknesses:**

Minor comments:
- Page 1: Please state whether $\tilde{\Delta}$ and/or $g$ are assumed to be known. From Eq.(3) it seems that we do assume that $\Delta$ is known.
- Page 2, line 82: The notation for $f_t:\mathbb{R}^N\times \mathbb{N} \to\mathbb{R}^N$ is confusing/informal because from the displayed equation, it seems that $f_t$ takes input from $\mathbb{R}^N$. Is $t$ supposed to be the input integer in $\mathbb{N}$?
- Page 2, line 83: If $f_t^a$ are general Lipschitz functions, it is not clear to me why $f_t$ is linear.
- Page 3, line 112: It seems to me that if we replace the second moment assumption by finite $k$-th moment, we are strengthening the assumption, not weakening it.

**Questions:**

- In the abstract it is claimed that the spectral method is shown to match the information-theoretic transition. If I understand the arguments of the paper correctly, it seems that this is only a conjecture that is based on a heuristic, and no rigorous proof was given. Did I miss anything?

**Limitations:**

No concerns regarding potential societal impact of this work.

---

> ### Author Rebuttal · Authors · 2023-08-09
>
>
> We express our gratitude to the reviewer for their insightful comments and valuable suggestions. We will incorporate all the suggestions into the final version, regardless of acceptance.
>
> Weakness, Page 1: Indeed the matrix $\tilde{\Delta}$ and the partition function $g$ are assumed to be known. Note however, that this does not have to be a limitation.
>
> In the degree-corrected block model Karrer and Newman (2011), for instance, the noise profile is perfectly known and given by the degrees of nodes in the graph. Additionally, for practical applications, one can empirically estimate the variance profile and assign group membership according to the empirical variances. We can first estimate both $\tilde{\Delta}$ and $g$ without knowing either of them in advance. Then we can work with the estimators rather than with the true quantities. We did not explore this any further in this work. An interesting direction for future work would be to see how AMP which uses estimators instead of the true $\tilde{\Delta}$ and $g$ performs and quantify how the lack of information impacts the performance of the algorithm, for instance using the Expectation Maximization framework in AMP (e.g. as in Expectation-Maximization Gaussian-Mixture Approximate Message Passing,  Vila \& Schniter 2013)
>
> Weakness, Line 82: It is true that the definition is a bit confusing. We meant $t$ to be an integer and for each $t \in \mathbb{N}$ $f_{t}$ is a function $\mathbb{R}^{N} \rightarrow \mathbb{R}^{N}$.
>
> Weakness, Line 83: The word linear is a mistake here and should be left out.
>
> Weakness, Line 112: You are right, the right word is strengthen.
>
> Question: This is indeed correct and we will clarify this. The method we use to show the weak recovery at the sharp transition used a heuristic argument from State Evolution/AMP theory. While the argument is mathematically solid (see e.g. Maillard, Krzakala, Lu and Zdeborova [2021], or Venkataramanan, Kogler, and Mondelli [2022]) and backed by numerical simulations, it is not fully rigorous. A direct rigorous treatment of the spectral method directly from random matrix theory is a challenging but interesting open problem in random matrix theory.

---

> > ### Comment · Reviewer_kH5e · 2023-08-16
> > **Reply to authors' rebuttal**
> >
> > I would like to thank the authors for their reply. My assessment of the paper remains positive.

---

### Official Review · Reviewer_R9KJ · 2023-07-11

**Soundness:** 4 excellent
**Presentation:** 3 good
**Contribution:** 3 good
**Rating:** 6
**Confidence:** 4

**Summary:**

This paper considers the spiked Wigner problem with inhomogeneous noise, i.e. the inverse problem of estimating a rank-matrix through an inhomogeneous noise channel. This problem naturally arises in many applications and a universality result makes the problem considered quite general with regards to the noise distribution.

The authors have made several contributions in this paper:
1. They have derived the AMP recursions to solve the spiked Wigner problem.
2. The most interesting property of AMP-like methods is that their behavior can be characterized exactly through a set of low-dimensional state evolution equations. This paper obtains the state evolution for the AMP recursion that solves the problem considered.
3. The authors analyze the AMP algorithm with identity denoisers and show that it corresponds to an spectral method for a specific matrix. More interestingly, the authors conjecture that this linear version of AMP detects a spike in the same region as the general AMP.

**Strengths:**

- The paper considers an interesting problem with many applications.
- The authors derive an AMP method to solve this problem.
- The authors obtain the state evolution of this AMP recursion, thus fully characterizing the macroscopic behavior of the AMP recursion at **each** iteration of the algorithm and **not** just the fixed points. This result gives us a **theoretical** way of obtaining the estimation error of the AMP method using very general metrics
- The authors show that when the Bayes optimal denoisers are used (i.e. the mean of the posterior) the fixed point equations exactly match the Bayes optimal fixed point equation of another recent work.
- Finally, the authors analyze the linear version of the AMP algorithm and show it is equivalent to spectral method for a specific matrix.

To summarize, the authors look at the spiked Wigner problem with inhomogeneous noise, derive an AMP method to solve it, and fully characterize the theoretical behavior of the AMP method in a certain high-dimensional asymptotics. This is in contrast to many other methods that are used in practice, but have no theoretical guarantees or guarantees of the form of high-probability upper bounds on the error in certain metrics. AMP methods, allow us to obtain the **exact** error in many **different error metrics** in a certain **high-dimensional limit**.

**Weaknesses:**

The weaknesses that come to my mind are mostly the usual weaknesses of the AMP algorithms:
- The AMP algorithms are often described as not very useful to solve problems in practice due to their instability often requiring a lot of tweaks such as damping to make them converge. I do not see much comments in this work regarding the stability of the AMP method described. This is in part due to generality of the algorithm (described using general denoisers) and in fact for the linear case the authors show a condition for convergence, however no mention of convergence or potential issues in the general case is mentioned. I should admit however that for example Bayes optimality of the AMP method with Bayes optimal denoisers makes this method interesting in practice for this problem.
- That being said, AMP having theoretical guarantees would still be very valuable as a theoretical tool. However, obtaining the errors through the state evolution equations are often very nontrivial due to the need to calculate expectations in a recursive formula that often need MCMC methods and are sometimes computationally not much better than running AMP for several instances of the problem and estimating the final error.
- As mentioned above, the AMP results only hold in a certain asymptotic regime. However in practice, very good match is observed even for moderately sized problems as the authors also mention.

## Minor comments
I believe the work is hard to follow for someone who is even familiar with the problem considered but not familiar with the AMP literature. It assumes the readers are knowledgeable in this area and very familiar with previous works which makes it hard to follow.

**Questions:**

- It would be nice if more explanations are added regarding the convergence of the AMP algorithm, any issues it might have in practice and the usefulness of the state evolution equations as they are recursive equations that do not seem trivial to simulate. For example, the convergence of AMP, or uniqueness of its fixed points (if it is in fact unique) should either be shown or it should be clearly mentioned that these are open questions that need to be addressed in order for this AMP method to be fully characterized.
- In many AMP papers, the authors make a correspondence between the AMP algorithms and estimators that are more familiar to the community. For example, using certain (generalized) proximal operators as the denoisers, the AMP would then can be shown to have fixed points that are also critical points of a certain loss function, i.e. AMP would be doing a form of M-estimation. Similarly, one could make a correspondence between other Bayesian estimators such as MMSE estimator and denoisers that use the posterior distribution such as the Bayes optimal one also mentioned in the paper. Can such results be shown for the AMP algorithm considered here? For example showing that AMP with a certain denoiser is minimizing a certain loss function, etc.? Since people are much more familiar these optimization based estimators, it would make this paper a lot easier to understand and follow. Similarly for the Bayesian estimators which are trying to estimate the signal based on the posterior, such as MMSE estimator, it would be good to make the correspondence between AMP and such estimators more clear.

**Limitations:**

- The practical limitations of the AMP algorithms and how easy/hard it is to use the state evolution in practice are not adequately discussed.

---

> ### Author Rebuttal · Authors · 2023-08-09
>
> We express our gratitude to the reviewer for their insightful comments and valuable suggestions. We will incorporate all the suggestions into the final version, regardless of acceptance.
>
> • Weaknesses: We acknowledge that we did not delve into potential stability issues or convergence problems that our AMP algorithm might encounter. Such challenges are indeed common to many similar algorithms and addressing them lies beyond the scope of our current work. Nevertheless, in our specific case, the AMP is Bayes-optimal, which typically exhibits strong convergence properties.
>
> Additionally, while we recognize these concerns, they are precisely the reason we introduced the (AMP-inspired) spectral method in Section 3. This method can be formulated and solved using an off-the-shelf Python solver in just a few lines. We wish to highlight that this spectral algorithm achieves performance close to that of AMP without any associated algorithmic issues.
>
> In response to the second point, we concur and plan to comment on it in an updated version.
>
> • Questions: To begin with, our AMP does indeed provide the Bayes-optimal MMSE performance by computing the posterior mean. From the Bayesian statistics perspective, this is a classical estimator, and we will elucidate this in the revised version.
>
> Furthermore, our AMP and its state evolution can be seamlessly adapted to examine M-estimators for this challenge, in the same vein as the standard AMP. Here, the denoiser will simply transition to the corresponding convex proximal operator. This is well-documented in sources such as "Generalized Approximate Message Passing for Estimation with Random Linear Mixing" by Rangan (2012) and "High Dimensional Robust M-Estimation: Asymptotic Variance via Approximate Message Passing" by Montanari and Donoho (2013). For M-estimation, our strategy in section 3 pertaining to the spectral method can be perceived in a similar light. We modify the AMP to develop a spectral method, which could also be interpreted as minimizing a tailored M-estimator. We will provide clarity on this aspect in the forthcoming version.

---

### Decision · Program_Chairs · 2023-09-21

**Decision:**

Accept (poster)

**Comment:**

While there were some initial concerns about the balance between technical content and more accessible material, these were generally alleviated following the rebuttal, and no major concerns remain.  The consensus is acceptance, though the reviewer’s comments and suggestions should still be carefully incorporated into the camera-ready versions.